# Seismic Response Analysis of Reinforced Concrete Frame Structures Considering Slope Effects

Pengyan Song, Shuang Guo, Wenao Zhao and Qin Xin *

College of Civil Engineering and Architecture, Hebei University, Baoding 071002, China;
songpengyan@sina.com (P.S.); guoshuang0317@sina.com (S.G.); z18332397516@sina.com (W.Z.)
* Correspondence: xinqin0902@163.com

**Abstract:** According to the seismic damage due to past events, buildings located on slopes can present a worse seismic performance. To explore this, this study established a finite element model based on a 6-story RC frame structure and soil models based on a practical slope using OpenSees software. Combining the superstructure model with the soil model through soil spring elements, three soil-structure interaction systems with different slope rates were set up. Twenty near-field seismic actions were used as input loads for dynamic time–history analysis. The analysis shows that in the process of seismic action, the deformation tendency of the structure is affected by the slope. There is a clear tendency for lateral displacement towards the slope, and it is more obvious with a greater slope ratio. Meanwhile, the slope has no impact on the shear force at the base of the structure or at the bottom of the column. In addition, there is no correlation between the degree of impact and the slope gradient on the peak value of internal forces and deformations of structure.

**Keywords:** reinforced concrete structure; slope; soil–structure interaction; seismic response

## 1. Introduction

Soil–structure interaction (SSI) is an important research direction in the seismic research field. A lot of buildings in many countries around the world are located on slopes. According to the seismic damage due to past events, buildings located on slopes can present a worse seismic performance. Therefore, the effect of the slope should be considered when studying the impact of SSI on the seismic performance of structures [1].

Many scholars have paid attention to the dynamic soil–structure interaction and achieved remarkable results. Gu et al. [2] proposed an analytical method of coupling a numerical solution with an analytical solution, and used powerful finite element software to simulate the nonlinear behavior of complex structures to seek the solution. Zhao et al. [3] proposed a highly efficient analysis method for deep soil–structure interaction under earthquake activity, which improved the computational efficiency by reducing the size of the soil–structure interaction model. Zhang et al. [4,5] explored the generation and development processes of natural frequency, vertex displacement, and plastic hinge of the structure under the two assumptions of a rigid foundation and considering soil–structure interaction, and revealed the importance of considering the SSI effect when performing pushover analysis on the structure. They carried out low-cyclic reversed loading tests on an independent foundation-frame substructure, and analyzed the seismic performance of the structure considering the effect of soil. Mohammed El Hoseny et al. [6] demonstrated that the SSI effect plays a role in amplifying the lateral deflection of frame structures through experiments and numerical simulations. Paraskevi K. Askouni and Dimitris L. Karabalis [7] conducted a series of seismic analyses of asymmetric small low-rise three-dimensional reinforced concrete (R/C) buildings while deliberating the influence of deformable soil on the seismic structural response.

Many scholars have also focused on the effect of slope on the anti-seismic ability of the structure. Yan et al. [8] studied the deformation of the slope foundation, the upper structure,

and the raft under the combined action. Li et al. [9] analyzed the influence of slope height and slope angle on the slope section's response spectrum and spectrum ratio. They proposed the ground motion amplification factors of the rocky slope. Liu et al. [10] established a simplified analytical model of a building structure with an unequal height grounding slope and analyzed the internal force of this structure under the change of beam span and column height. Fan et al. [11] analyzed in detail the differences between buildings on a slope and on flat land in the following aspects: dynamic structural characteristics, lateral displacement and story drift ratio under horizontal ground motion, torsional displacement ratio considering accidental eccentric action, lateral stiffness ratio, and internal force of the structure. Zhang et al. [12] studied the dynamic displacements and internal forces of single-span and multi-span frame structures on deep pit slopes using a large-scale shaking table test. Rahul Ghosh and Rama Debbarma [13] utilized methods such as a nonlinear time history method (NLTHM) to study the effect of slope angle variation for the structures resting on sloping ground and revealed the importance of considering SSI in seismic analysis. Considering different slope angles and structure heights, Mohammad Javad Shabani et al. [14] investigated the seismic performance of three groups of moment-resistant frame (MRF) steel structures with 5, 10, and 15 stories through the three-dimensional (3D) numerical simulations. Results show that the topographic irregularities magnify the acceleration at a distance of 2–3 times the slope height (H) from the slope crest.

Although many scholars have paid attention to the study of soil–structure interaction, few scholars have focused on the impact of slope on seismic performance. In recent years, there have been few research results on slope–structure seismic performance. In view of this, this study took a practical slope based in Yunnan Province, China as the research background, and a 6-story and 3-span reinforced concrete frame structure as the research object. Then the finite element models of the upper structure and slopes were established and connected through soil springs from OpenSees. Finally, three soil–structure interaction systems with different slope rates were set up. The input seismic load was 20 near-field seismic wave records. By employing the time history analysis method, the influence of slope ratio on the superstructure, pile foundation, and soil mass were investigated.

## 2. Establishment of Soil–Structure Interaction Model

### 2.1. Determination of Boundary Condition

In general, the calculation results become more accurate with an increase in the calculation range. But this multiplies the complexity of the computation. When considering two-dimensional geotechnical problems in OpenSees, the accuracy and convergence can be satisfied by using the remote ordinary artificial boundary.

The distance between the soil's boundary and the structure foundation's edge is three times the width of the foundation. It can not only effectively control the calculation error of the model but also significantly improve the efficiency of the calculation. Therefore, the right pile of this structure is 45 m away from the right boundary, and the horizontal distance from the left pile to the left boundary is 45 m as well. This meets the above requirements. (Figure 1 shows the range of foundation soil.)

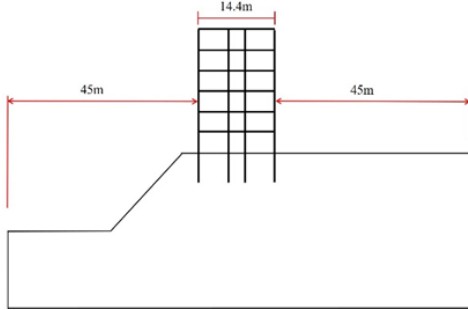

**Figure 1.** Schematic diagram of soil range.

### 2.2. Consideration of Pile–Soil Interaction

Piles connect the soil and superstructure. The connection between the soil and the pile was assumed to be rigid to simplify the calculation used in previous studies so that the relative slip of the pile and soil could be ignored. Motivated to avoid the error caused by this connection mode, the study used the pile–soil spring to connect the soil and the pile for simulation. In the finite element model, the pile and soil body were connected by zero-length nonlinear soil springs, namely *p-y* spring (lateral resistance) [15]. (Figure 2 is the schematic diagram of the soil spring model.)

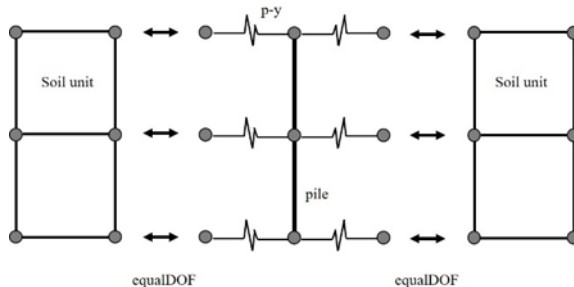

**Figure 2.** Schematic diagram of soil spring.

For clay, the study used Formulas (1)–(3) to express the *p-y* spring [16].

$$p_{ult} = c_u B N_p \tag{1}$$

$$N_p = \left( 3 + \frac{\gamma' x}{c_u} + \frac{J_x}{B} \right) \leq 9 \tag{2}$$

$$y_{50} = 2.5 B \varepsilon_{50} \tag{3}$$

where $P_{ult}$ is the ultimate resistance of the soil; $B$ is the pile diameter; $Np$ is the lateral bearing capacity coefficient; $c_u$ is the undrained shear strength of the soil; $\gamma'$ is the effective weight of the soil, $x$ is the depth of the soil; $\varepsilon_{50}$ is the strain corresponding to the ultimate stress of 50% stress in the stress–strain curve of the soil, and the value is taken as 0.005; $y_{50}$ is the pile deformation when the soil resistance reaches half the ultimate bearing capacity; and $J$ is the dimensionless coefficient, designated 0.5.

### 2.3. Establishment of a Reinforced Concrete Frame Structure Model Considering Soil–Structure Interaction

According to Chinese current norms [17], a reinforced concrete frame structure with six layers and three spans was designed. Three different slope ratios of 1:1, 1:2, and no slope were considered, respectively, in the design of the slope foundation.

Fortification intensity (planned basic acceleration) was VIII degree (0.2 g). The ground motion group was group 2. The site classification was class II. The primary reinforcement bar grade of the beam and column was HRB335. The stirrup grade was HPB235. The concrete grade was C35. The X-direction dimension of the soil was 104.4 m. The slope height was 18 m. The ratios of the slope were 1:1 and 1:2, respectively. The horizontal distance from the superstructure to the top of the hill was 3 m. The superstructure side-span was 6 m, and the middle span was 2.4 m. The structure's bottom layer was 3.9 m, and the standard layer height was 3.3 m. The size of the beam section was 0.25 m × 0.5 m. The section size of the column and pile was 0.6 m × 0.6 m. The pile length was 6 m. Figure 3 is the structure diagram, and Figure 4 shows the reinforcement information of the beam section and column section.

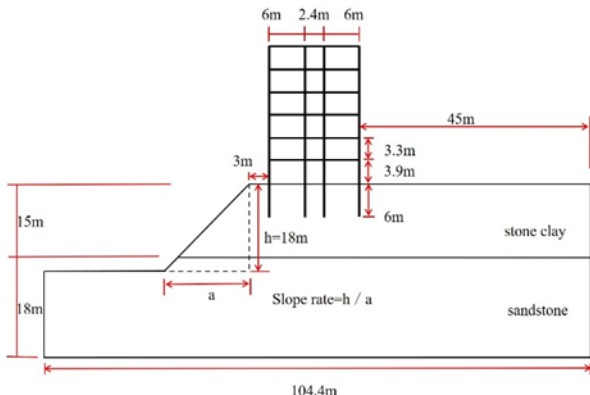

**Figure 3.** Structure diagram.

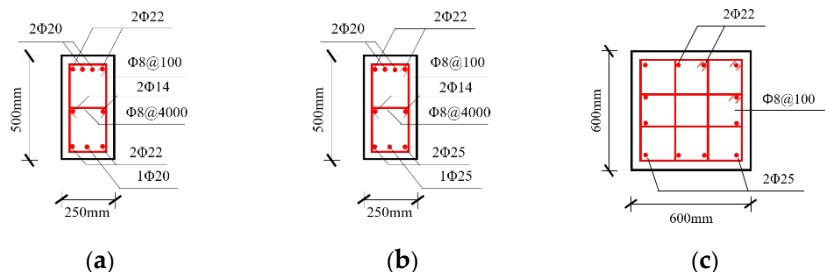

**Figure 4.** Section reinforcement drawing. (**a**) Side-span beam; (**b**) middle-span beam; (**c**) column.

This study used PIMY and PDMY from OpenSees to simulate the soil material. The two constitutive relations of the material were elastoplastic materials with multiple yield surfaces. Concrete01 from OpenSees was used to simulate the foundation beams, columns, and piles. The Steel02 constitutive model was adopted for longitudinal reinforcement in beams, columns, and piles. In addition, this study used PySimple1 and TzSimple1 materials from OpenSees to realize the characteristics of the *p-y* unit and the t-z unit in springs. Displacement-based beam-column elements were used to model beams, columns, and piles. The soil element adopted the four-node plane strain element. And the pile-soil interaction was simulated by the *p-y* zero-length element. Beams, columns, and piles were made of the same type of material. Tables 1 and 2 show the specific parameters of the materials of beams, columns, and piles. This research is based on actual slope engineering when considering soil parameters. Additionally, the property of each soil layer was assumed to be uniform. The details of the soil are provided in Table 3 [18]. The finite element model is shown in Figure 5.

**Table 1.** Concrete parameters.

| Structure Type | $f_c$/MPa | $\varepsilon_{co}$ | $f_{cu}$/MPa | $\varepsilon_{cu}$ |
|---|---|---|---|---|
| Non-Core Concrete | 29.76 | 0.002 | 0 | 0.004 |
| Core Concrete | 32.57 | 0.0022 | 20.76 | 0.0124 |

Where $f_c$ is concrete compressive strength (with positive value), $\varepsilon_{co}$ is concrete strain at maximum strength (with positive value), $f_{cu}$ is concrete crushing strength (with positive value), $\varepsilon_{cu}$ is concrete strain at crushing strength (with positive value).

**Table 2.** Rebar parameters.

| Structure Type | $f_y$/MPa | *E*/MPa |
|---|---|---|
| rebar | 388 | 200,000 |

Where $f_y$ is yield strength, *E* is initial elastic tangent.

**Table 3.** Soil parameters.

| Soil Type | $h$/m | $\gamma$/(kN/m³) | $\rho$/(ton/m³) | $E$/MPa | $v$ | $G_r$/MPa | $B_r$/MPa | $c$/kPa | $\varphi$ |
|---|---|---|---|---|---|---|---|---|---|
| Stone clay | 15 | 20 | 1.9 | 50 | 0.35 | 18.52 | 55.56 | 20 | 14.5 |
| Sandstone | 18 | 24.6 | 2.1 | 9500 | 0.25 | 3800 | 6333 | 50 | 42 |

Where $h$ is the soil depth; $\gamma$ is the gravity of the soil; $\rho$ is mass density of saturated soil; $E$ is the elastic modulus; $v$ is Poisson's ratio; $G_r$ is referenced low-strain shear modulus; $B_r$ is reference bulk modulus; $c$ is apparent cohesion at zero effective confinement; $\varphi$ is friction angle at peak shear strength in degrees, optional.

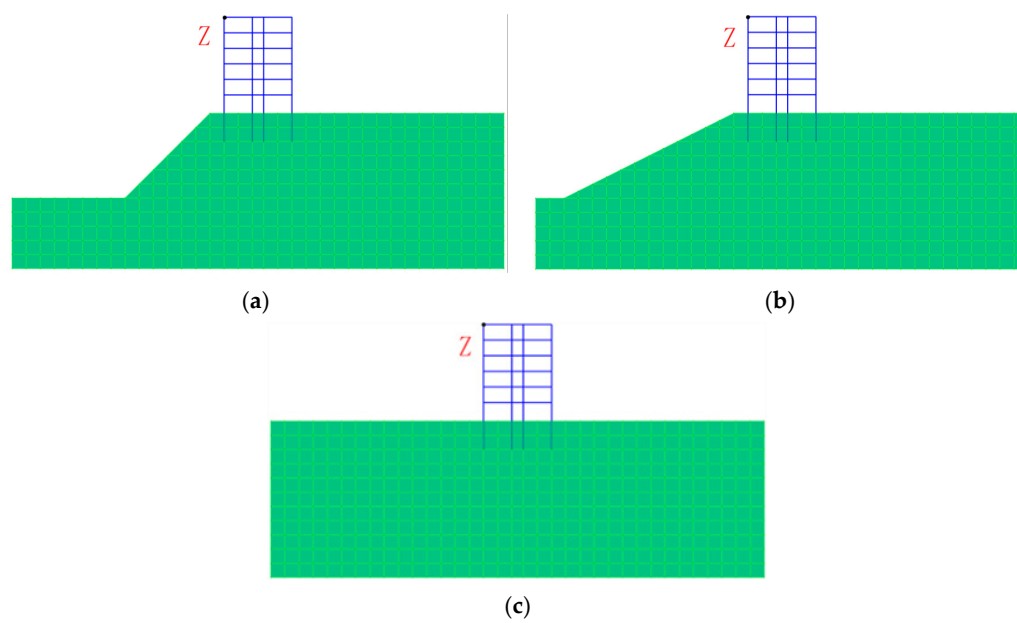

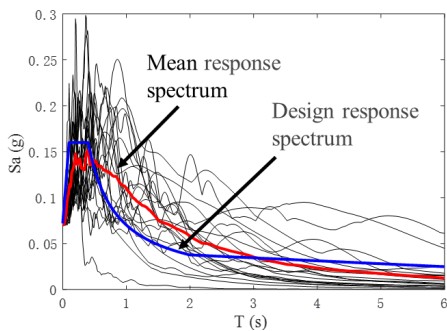

**Figure 5.** Finite element model. (**a**) 1:1; (**b**) 1:2; (**c**) no slope. The green part represents the soil unit, the blue part represents the upper structure, and the red font represents the observation points.

## 3. Analysis of Structural Response under Near-Field Earthquake

### 3.1. Select Seismic Wave

The near-field earthquake will strongly lash the structure due to its characteristics of a long period and velocity pulse [19]. Therefore, the authors selected 20 near-site seismic waves in the PEER database based on the seismic acceleration design response spectrum of the actual slope site. The selection principles [20] were as follows: (1) fault distance <20 km; (2) magnitude >6. Table 4 shows the information of the selected 20 near-field vibration records. The purpose of this study was to perform an analysis of the structural response to frequent earthquakes. So that the peak acceleration of the seismic waves was adjusted to 0.07 g. Figure 6 shows the response spectrum of the input wave. The average response spectrum of the 20 earthquake waves was similar to the design response spectrum of frequent earthquakes.

**Figure 6.** Response spectrum of input seismic wave.

**Table 4.** Near-site vibration records.

| Condition | Name | Observation Station | Moment Magnitude | Fault Distance/km |
|---|---|---|---|---|
| 1 | "Imperial Valley-06" | "El Centro-Meloland Geot. Array" | 6.53 | 0.07 |
| 2 | "Irpinia_ Italy-01" | "Sturno (STN)" | 6.9 | 10.84 |
| 3 | "Superstition Hills-02" | "Kornbloom Road (temp)" | 6.54 | 18.48 |
| 4 | "Loma Prieta" | "Gilroy-Historic Bldg." | 6.93 | 10.97 |
| 5 | "Loma Prieta" | "Saratoga-Aloha Ave" | 6.93 | 8.5 |
| 6 | "Cape Mendocino" | "Petrolia" | 7.01 | 8.18 |
| 7 | "Northridge-01" | "Newhall-W Pico Canyon Rd." | 6.69 | 5.48 |
| 8 | "Kocaeli_ Turkey" | "Yarimca" | 7.51 | 4.83 |
| 9 | "Chi-Chi_ Taiwan" | "CHY024" | 7.62 | 9.62 |
| 10 | "Chi-Chi_ Taiwan" | "TCU049" | 7.62 | 3.76 |
| 11 | "Chi-Chi_ Taiwan" | "TCU063" | 7.62 | 9.78 |
| 12 | "Denali_ Alaska" | "TAPS Pump Station #10" | 7.9 | 2.74 |
| 13 | "Cape Mendocino" | "Centerville Beach_ Naval Fac" | 7.01 | 18.31 |
| 14 | "Parkfield-02_ CA" | "PARKFIELD-EADES" | 6 | 2.85 |
| 15 | "Parkfield-02_ CA" | "Parkfield-Cholame 3E" | 6 | 5.55 |
| 16 | "Parkfield-02_ CA" | "Parkfield-Fault Zone 12" | 6 | 2.65 |
| 17 | "Montenegro_ Yugoslavia" | "Bar-Skupstina Opstine" | 7.1 | 6.98 |
| 18 | "L'Aquila_ Italy" | "L'Aquila-V. Aterno-Centro Valle" | 6.3 | 6.27 |
| 19 | "Chuetsu-oki_ Japan" | "Joetsu Kakizakiku Kakizaki" | 6.8 | 11.94 |
| 20 | "Darfield_ New Zealand" | "DSLC" | 7 | 8.46 |

### 3.2. Analysis of Structural Deformation

Table 5 indicates the peak displacement of the top floor and the maximum inter-story displacement angle of the structure under the action of 20 seismic waves.

**Table 5.** Crest value of displacement of top floor and maximum inter-story displacement angle.

| Condition | Top Displacement/mm | | | Maximum Inter-Story Displacement Angle | | |
|---|---|---|---|---|---|---|
| | 1:1 | 1:2 | No Slope | 1:1 | 1:2 | No Slope |
| 1 | 88.6431 | 87.2133 | 85.0686 | 0.005162 | 0.005030 | 0.005255 |
| 2 | 86.0306 | 81.3623 | 91.7378 | 0.005043 | 0.004846 | 0.005442 |
| 3 | 121.0670 | 119.0080 | 120.2630 | 0.007318 | 0.007256 | 0.007330 |
| 4 | 58.7790 | 59.8172 | 61.8797 | 0.004435 | 0.004453 | 0.004381 |
| 5 | 56.6783 | 55.3526 | 55.1922 | 0.003392 | 0.003394 | 0.003336 |
| 6 | 28.9906 | 28.2548 | 29.1686 | 0.002407 | 0.002300 | 0.002862 |
| 7 | 152.7920 | 150.1290 | 152.8640 | 0.009885 | 0.009820 | 0.009874 |
| 8 | 123.3040 | 120.5070 | 120.0740 | 0.007278 | 0.007158 | 0.007146 |
| 9 | 82.1381 | 81.4840 | 83.7088 | 0.005498 | 0.005493 | 0.005132 |
| 10 | 41.6379 | 42.7502 | 43.9247 | 0.003094 | 0.003042 | 0.003157 |
| 11 | 134.3620 | 131.1240 | 132.6730 | 0.008390 | 0.008265 | 0.008319 |
| 12 | 140.6770 | 141.5030 | 142.1080 | 0.008770 | 0.008711 | 0.008865 |
| 13 | 76.3399 | 76.3586 | 79.5788 | 0.004586 | 0.004561 | 0.004578 |
| 14 | 37.1906 | 36.2943 | 36.4235 | 0.002077 | 0.002050 | 0.002177 |
| 15 | 4.7977 | 3.8273 | 3.5885 | 0.000511 | 0.000516 | 0.000492 |
| 16 | 65.9042 | 64.7746 | 65.9322 | 0.004258 | 0.004242 | 0.004351 |
| 17 | 70.8350 | 69.2837 | 71.0549 | 0.003835 | 0.003752 | 0.004016 |
| 18 | 21.2565 | 20.5659 | 18.8895 | 0.001481 | 0.001514 | 0.001548 |
| 19 | 75.7871 | 73.4891 | 78.4525 | 0.004145 | 0.004136 | 0.004167 |
| 20 | 38.3799 | 38.7754 | 42.5813 | 0.002723 | 0.002794 | 0.002685 |

As shown in Table 5, the peak displacement of the top floor of the structure differed under different ground motions, as did the maximum inter-story displacement angle of the structure. It is not clear how the slope affected the peak value of the top floor displacement and the maximum inter-story displacement angle. The slope ratio had different effects

on the story drift ratio under condition 6 and condition 18. Therefore, this study mainly conducted a comparative analysis of the structural response under the two conditions.

Figure 7 shows, respectively, the time history curve of the earthquake wave's acceleration of conditions 6 and 18. Figure 8 shows the change curves of structures' story drift ratio with the change of floor height under conditions 6 and 18. It can be seen from the figures that the difference in ground motion records significantly influenced the story drift ratio of the structure. However, the change rules were similar. That is, as the stories increased, the story drift ratio decreased first, then increased, and finally decreased. The maximum inter-story displacement angle of the structure appeared on the fourth or fifth floor. Next, the impact of the slope on the inter-story displacement angle of the structure was analyzed based on the results of specific seismic actions.

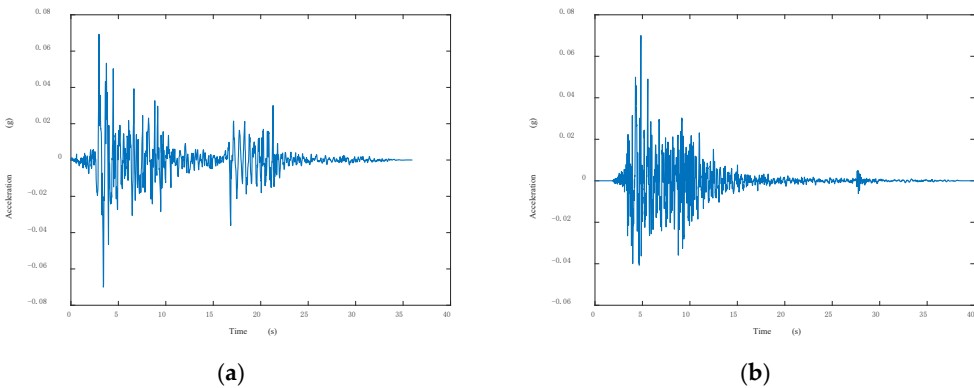

(**a**)         (**b**)

**Figure 7.** Time history curve of seismic wave acceleration: (**a**) working condition 6; (**b**) working condition 18.

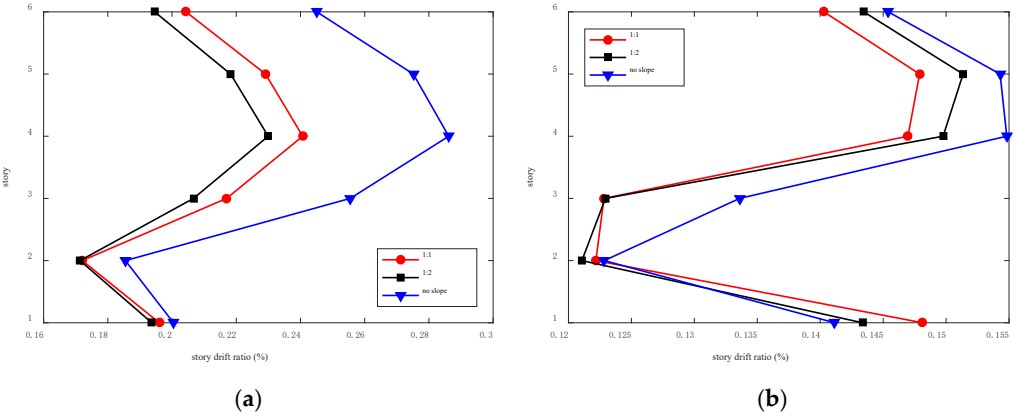

(**a**)         (**b**)

**Figure 8.** Inter-story displacement angle of each structure: (**a**) working condition 6; (**b**) working condition 18.

Under condition 6, the story drift ratio of the structure without the slope was the largest, and the story drift ratio of the structure with the 1:2 slope ratio was the smallest. In the fourth layer, the difference in the story drift ratio of the three structures was the most significant. Under condition 18, the slope ratio had different effects on the story drift ratio. The positions of the maximum inter-story displacement angle of the three structures were different. The maximum inter-story displacement angle of the no-slope structure occurred on the fourth floor, while that of the 1:1 slope ratio structure occurred on the first floor, and that of the 1:2 slope ratio structure occurred on the fifth floor. The maximum inter-story displacement angle of the structure decreased with the increase in the slope ratio. This phenomenon shows that there is no qualitative conclusion about the effect of slope ratio on the structure's story drift ratio.

The time history curve of the top floor displacement under condition 6 and condition 18 is represented in Figure 9. Point z is the monitor point (see Figure 5).

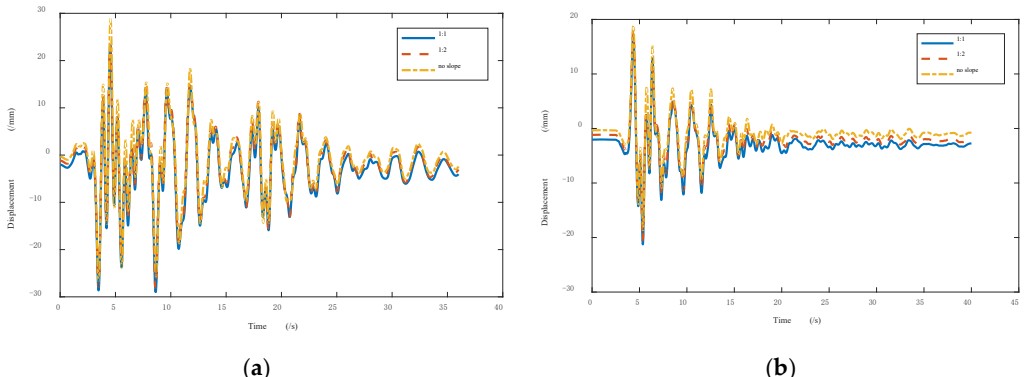

**Figure 9.** Time history curve of horizontal displacement of the top layer: (**a**) working condition 6; (**b**) working condition 18.

From Figure 9, it can be found that the time history curve of the top floor displacement varied with different vibrations. However, the difference of ground motion did not change the impact of slope on the displacement of the top layer of the structure. The slope made the horizontal displacement of the top layer show a negative X-direction trend, that is, the movement trend towards the slope direction. Additionally, the greater the slope of the soil, the more pronounced this trend was, while the impact of slope rate on the displacement of the top layer is apparent in working condition 18.

Figures 10 and 11, respectively, show the lateral displacement time history curve of the piles and soil. The monitor points were at buried depths of 3 m and 6 m of the leftmost pile. As seen in the figures, the lateral displacement of the pile was different under the action of the two seismic waves, as was the lateral displacement of the soil. Additionally, the slope rate had the same influence on the lateral displacement of piles and soil.

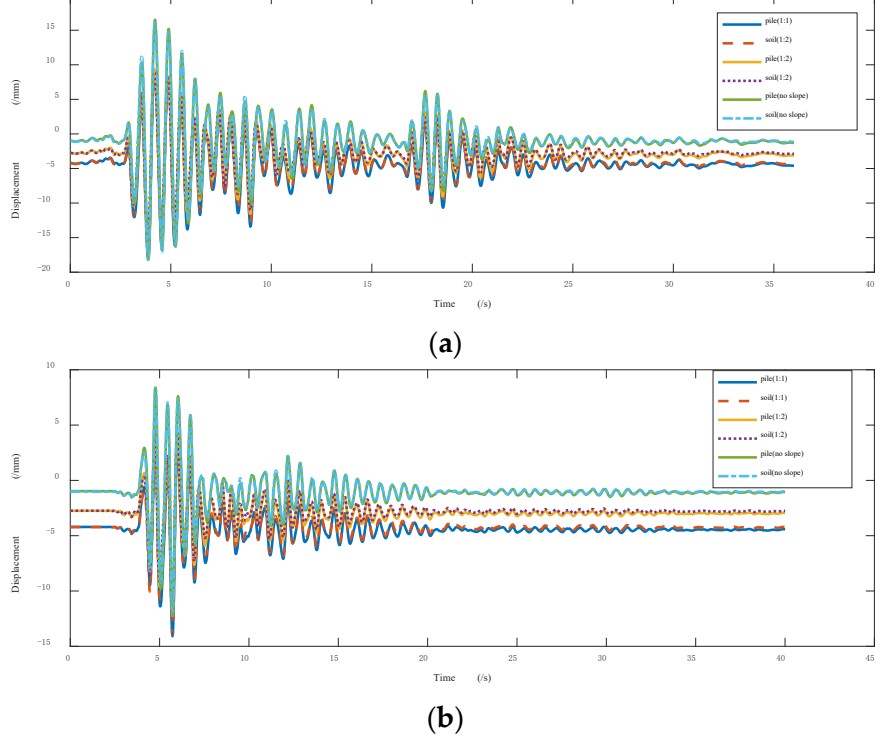

**Figure 10.** Displacement time history curve of pile and soil (at 3 m buried depth): (**a**) working condition 6; (**b**) working condition 18.

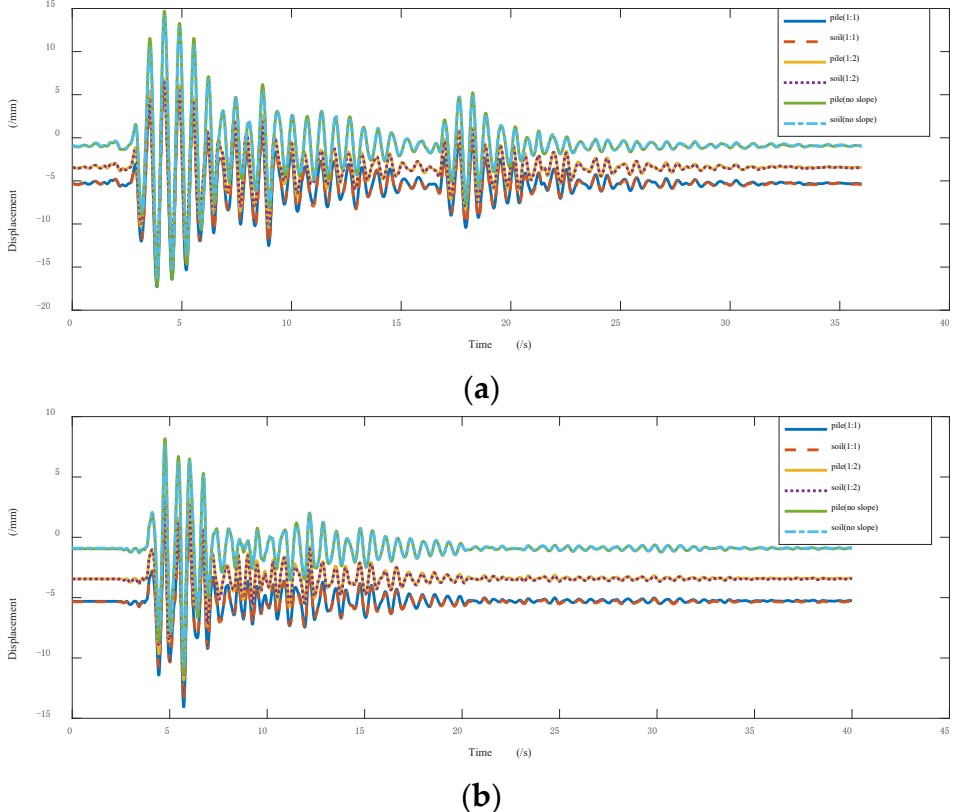

**Figure 11.** Displacement time history curve of pile and soil (at 6 m buried depth): (**a**) working condition 6; (**b**) working condition 18.

During the same seismic wave action, the change law of lateral displacement of piles buried at 3 m was similar to that of piles buried at 6 m. The law of lateral displacement changes of soil buried at 3 m was similar to that at 6 m. At the end of the action, the displacement of the pile and soil was almost the same, which shows that the soil spring used in the numerical model simulated the pile–soil interaction well.

At the burial depth of 3 m, the pile and soil had a small relative displacement. The lateral displacement of soil was slightly smaller than the horizontal displacement of the pile at the same depth. The pile and soil of the structure with the slope both had great lateral displacement towards the slope direction. This law was more obvious with the increase in slope rate. However, the pile and soil of the structure without slope had lateral displacement in the X and negative X directions. This phenomenon indicates that there was a clear tendency for lateral displacement of piles and soil towards the slope, and it was more significant with increasing slope ratio.

At the burial depth of 6 m, the displacement of the pile was almost the same as that of soil at the same time. The slope had a certain effect on the displacement of the pile and soil, and the law was roughly the same as that of the position buried at 3 m depth. As the depth increased, however, the change in slope rate had a more significant influence on the lateral displacement of the pile and soil, with a greater degree of dispersion. This points to a much more dramatic effect of the slope on the lateral displacement response of the pile and soil at the bottom of the structure.

### 3.3. Internal Force Analysis of the Structure

Table 6 shows the peak value of the base shear and the side column's base shear under the action of 20 seismic waves. Different ground motions significantly affected the peak value of the base shear and column bottom shear of the structures. However, the influence of the slope foundation on them was not obvious.

The plots in Figures 12 and 13, respectively, show the base shear and column bottom shear (leftmost column) of the three structures at each time under two different seismic waves. Figure 14 shows the base shear–top floor displacement curve. It can be seen from the figures that, under the action of seismic waves, the existence of the slope had a slight impact on the base shear. Similarly, the slope hardly affected the column bottom shear force, yet the shear at the bottom of the column had a positive X trend at the initial stage. This effect is more obvious with the increase of the slope rate. After the initial phase, the impact of the slope on the shear at the bottom of the column and the shear at the base were very small, which can be almost ignored.

**Table 6.** Peak value of base shear and column bottom shear.

| Condition | Base Shear/kN | | | Column's Base Shear/kN | | |
|---|---|---|---|---|---|---|
| | 1:1 | 1:2 | No Slope | 1:1 | 1:2 | No Slope |
| 1 | 258.1188 | 258.7085 | 270.6870 | 64.6294 | 65.8020 | 66.2971 |
| 2 | 276.6748 | 272.5098 | 288.7356 | 69.8582 | 69.5527 | 73.1098 |
| 3 | 314.7402 | 316.0948 | 312.2768 | 78.0602 | 79.0076 | 78.0788 |
| 4 | 245.8858 | 248.7526 | 248.5819 | 56.3924 | 57.0828 | 57.3976 |
| 5 | 179.6923 | 181.4757 | 181.2229 | 43.8218 | 44.6971 | 45.4954 |
| 6 | 116.1666 | 116.6909 | 121.2996 | 29.7034 | 30.2304 | 30.8673 |
| 7 | 373.3306 | 374.1047 | 373.2631 | 93.6985 | 94.2365 | 94.1401 |
| 8 | 318.9546 | 318.6235 | 317.0211 | 79.5174 | 79.8410 | 79.7661 |
| 9 | 286.0369 | 287.6453 | 278.1560 | 65.9058 | 66.8384 | 64.0351 |
| 10 | 181.5003 | 181.4574 | 192.2615 | 41.1669 | 41.3942 | 43.9225 |
| 11 | 342.9019 | 343.3873 | 343.1056 | 85.0922 | 85.7891 | 86.1147 |
| 12 | 370.4145 | 370.2473 | 369.8107 | 92.6915 | 93.0185 | 93.0916 |
| 13 | 252.7692 | 255.8633 | 250.0474 | 58.3872 | 59.0756 | 57.9525 |
| 14 | 102.1527 | 103.4188 | 114.7740 | 25.9615 | 27.8515 | 28.8789 |
| 15 | 22.6516 | 24.6531 | 20.2570 | 7.5895 | 7.5382 | 4.6475 |
| 16 | 216.2035 | 217.0157 | 222.7302 | 52.3521 | 52.7976 | 55.4414 |
| 17 | 175.3438 | 175.3393 | 192.8594 | 42.4570 | 43.8822 | 48.1642 |
| 18 | 105.0791 | 111.9661 | 101.1974 | 27.9771 | 30.7484 | 25.6760 |
| 19 | 237.8861 | 242.5412 | 232.6821 | 59.9351 | 62.1678 | 58.9211 |
| 20 | 175.6080 | 182.3061 | 160.3153 | 45.3386 | 47.8601 | 40.9953 |

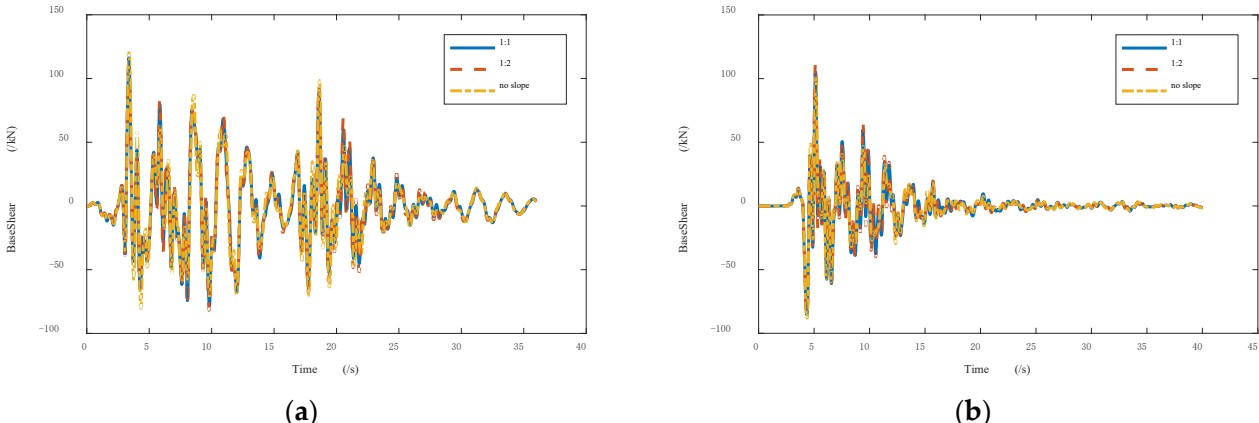

(**a**)          (**b**)

**Figure 12.** Time history curve of base shear: (**a**) working condition 6; (**b**) working condition 18.

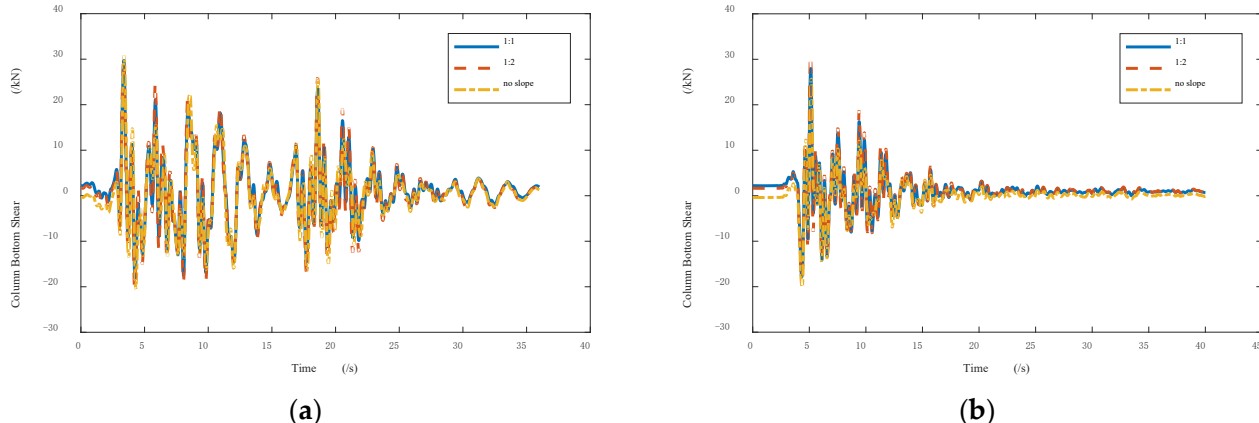

**Figure 13.** Time history curve of column bottom shear force: (**a**) working condition 6; (**b**) working condition 18.

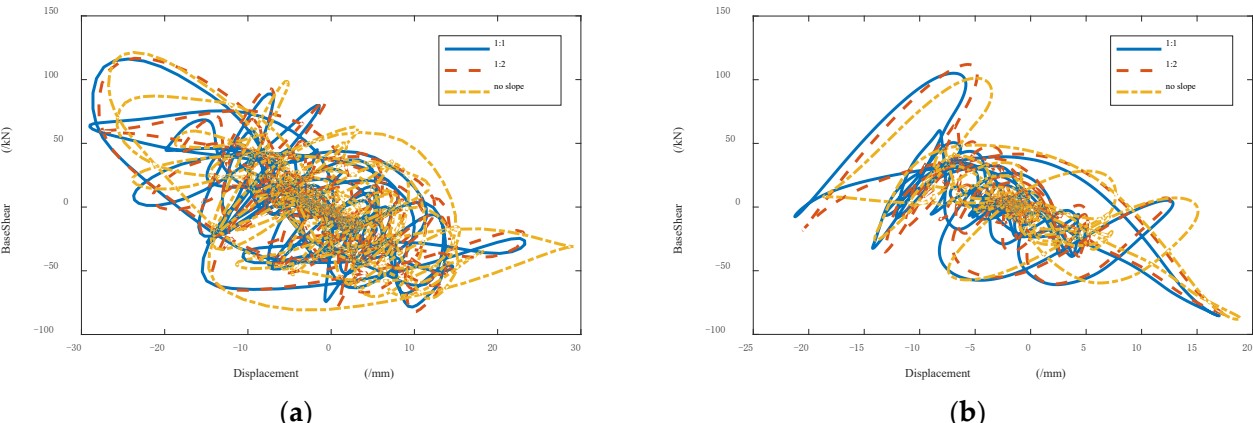

**Figure 14.** Base shear- top displacement hysteretic curve: (**a**) working condition 6; (**b**) working condition 18.

### 4. Conclusions

This study aimed to explore the effect of slope on the seismic performance of a structure. Therefore, this study took a practical slope as the research background, and a 6-story and 3-span reinforced concrete frame structure as the research object. Then, the finite element model of three slope structure interaction systems with different slope ratios was established using OpenSees, and 20 near-field seismic wave records were taken as input seismic load to perform kinetic time interval analysis and compute the dynamic response. Finally, this study explored the influence of slope ratio on the deformation and internal forces of the upper structure and the deformation of piles and soil according to numerical results. Through the analysis of this research result, the following conclusions are obtained.

1. Under the action of different seismic waves, the influence of slope rate on the story drift ratio is quite different. There is no qualitative conclusion about the effect of the slope ratio on the structure's story drift ratio.
2. Unlike the influence of the slope rate on the story drift ratio, however, under different seismic waves, the slope rate has the same effect on the top layer displacement. That is, the slope makes the horizontal displacement of the top layer show a negative X-direction trend, namely, the top layer moves toward the slope direction, and the higher the slope rate, the more pronounced this effect is.
3. The slope rate has the same effect on the lateral displacement of the pile as on the top layer displacement. The lateral displacement of the soil follows the same pattern.

4. The shear at the bottom of the column has a positive X trend at the initial stage. This effect is more obvious with an increase in the slope rate. After the initial phase, the impact of the slope on the shear at the bottom of the column and the shear at the base is very small, which can be almost ignored.

5. There is no correlation between the degree of impact and the slope gradient on the peak value of internal forces and deformations of the structure.

From the above conclusion, it can be seen that the trend of structural deformation is significantly influenced by slope, while the effect of slope on the peak value of deformation and force is influenced by the difference of seismic motions, and no clear conclusion can be drawn. In the seismic design of sloping buildings, structural response under sufficient seismic action should be analyzed and demonstrated.

**Author Contributions:** Methodology, P.S.; Software, P.S. and S.G.; Data curation, W.Z.; Writing—original draft, S.G.; Writing—review & editing, Q.X. All authors have read and agreed to the published version of the manuscript.

**Funding:** This research was funded by [the Institute of Engineering Mechanics, China Earthquake Administration] grant number [2019EEEVL0202], [Advanced Talents Incubation Program of Hebei University] grant number [521000981082] and [Science and Technology Project of Hebei Education Department] grant number [BJ2019042].

**Data Availability Statement:** The data presented in this study are available on request from the corresponding author. The data are not publicly available due to [the privacy of the data].

**Acknowledgments:** The present work has been supported by the Institute of Engineering Mechanics, China Earthquake Administration (2019EEEVL0202), Advanced Talents Incubation Program of Hebei University (grant No. 521000981082), and Science and Technology Project of Hebei Education Department (BJ2019042). These supports are gratefully acknowledged.

**Conflicts of Interest:** The authors declare no conflict of interest.

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
