# Peer review of "Seismic Response Analysis of Reinforced Concrete Frame Structures Considering Slope Effects"

_applsci, doi:10.3390/app13085149_

Round 1

Reviewer 1 Report

The manuscript (entitled: Seismic response analysis of reinforced concrete frame structures considering slope effects) pays main attention to the soil-structure interaction. To explore it, the study established the FEM model by Open software that was based on a 6-story RC frame structure. By combining the FEM model with the multi-yield surface model, three soil-structure interaction systems with different slope rates were set up. The analysis shows that in the process of seismic action, the deformation tendency of the structure is affected by the slope.

Author Response

Please refer to the latest attachment, which contains comprehensive content.

Reviewer 2 Report

  1. How does soil-structure interaction (SSI) affect the seismic performance of buildings? What is the role of slope in SSI and why is it important to study? How was the FEM model for the reinforced concrete frame structure and the multi-yield surface model based on the plane four-node element established?
  2. What were the input seismic loads used in the numerical simulation analysis? What was the effect of slope on the deformation tendency of the structure during seismic action? Is there a clear tendency of lateral displacement towards the slope and does it vary with the slope ratio?
  3. What is the impact of slope on the shear force at the base of the structure and the shear force at the bottom of the column? Does the peak displacement of the top floor of the structure differ under different ground motions and if so, how?
  4. How does the slope rate impact the peak value of inter-story displacement angle, base shear, and column bottom shear? Is there a correlation between the degree of impact and the slope gradient on the peak value of internal forces and deformations of structure?
  5. What is the bedding clay slope used in the study and why was it selected as the research background? How was the finite element model of slope-structure interaction established using OpenSees?
  6. What were the 20 near-field seismic wave records used in the study and why were they selected? What is the effect of slope ratio on the superstructure, pile foundation, and soil mass according to the numerical results?
  7. Is there a qualitative conclusion about the effect of slope ratio on the structure's story drift ratio? How does the slope affect the top layer displacement and is there a correlation with the slope rate?
  8. What is the impact of the slope rate on the lateral displacement of the pile and the soil's lateral displacement? How does the slope affect the shear at the bottom of the column and is there a correlation with the slope rate? What are the implications of the research results for seismic design of slope buildings and relevant codes?

Minor editing of English language required.

Reviewer 3 Report

Dear authors

What I am missing is the justification for case 6 and 18 detailed analysing only. Are they representative for the issue? Are they the most interesting?

As I understand correctly the table 5 and the figures are the results of your calculations. Please state it clearly.

I would like to draw your attention that China Is not the only country with buildings located on slopes. I am missing discussion on other than Chinese articles treating  on the issue in Chapter 1. [e. g https://link.springer.com/article/10.1007/s40091-019-0219-3]

Why did you enclose condition 14 case for analysis. That is against your selection principles (Magnitude >6). The moment magnitude in the above case is only 1,37.

Line 118 – you mention about “our current norms”. Non Chinese reader has no idea what norms you are having on your mind. Please add them to the References section

Please check the concrete class (line 124) shouldn’t it be C35/45?

The article needs some English proof checking [e. g. - So are the peak value of the inter-story….(line292)]

Editorial remarks:

1.       Dot missing

a.      Line 22 [shear / While]

b.      Line 249

c.      Line 96

d.      Line 106

2.       Table numbering – there are two Table 3 and table 4 is missing.

3.       Line 168 – adjusting

4.       Caption for Table 5 is on the other page than the table itself.

5.       Caption for Figure 13 is on the other page than the figure itself.

Please check the whole article but special attention must be paid at lines 291-296, 297-300

Round 2

Reviewer 2 Report

the revised manuscript can be accepted. 

Minor editing of English language is required.